# The effect of body image dissatisfaction on goal-directed decision making in a population marked by negative appearance beliefs and disordered eating

Jakub Onysk[1,2], Peggy Seriès[1]*

1 Institute for Adaptive and Neural Computation, School of Informatics, University of Edinburgh, Edinburgh, United Kingdom, 2 Max Planck Centre for Computational Psychiatry and Ageing Research, University College London, London, United Kingdom

* pseries@inf.ed.ac.uk

**Data Availability Statement:** Decision-making task behavioural data, questionnaire scores, and fitted

## Abstract

Eating disorders are associated with one of the highest mortality rates among all mental disorders, yet there is very little research about them within the newly emerging and promising field of computational psychiatry. As such, we focus on investigating a previously unexplored, yet core aspect of eating disorders–body image dissatisfaction. We continue a freshly opened debate about model-based learning and its trade-off against model-free learning–a proxy for goal-directed and habitual behaviour. We perform a behavioural study that utilises a two-step decision-making task and a reinforcement learning model to understand the effect of body image dissatisfaction on model-based learning in a population characterised by high scores of disordered eating and negative appearance beliefs, as recruited using Prolific. We find a significantly reduced model-based contribution in the body image dissatisfaction task condition in the population of interest as compared to a healthy control. This finding suggests general deficits in deliberate control in this population, leading to habitual, compulsive-like behaviours (body checking) dominating the experience. Importantly, the results may inform treatment approaches, which could focus on enhancing the reliance on goal-directed decision making to help cope with unwanted behaviours.

## Introduction

Eating disorders (ED) form a group of different conditions that, in a detrimental way affects a person's relationship with food. This usually leads to physical and psychological problems, which severely decrease the quality of life. In fact, these problems result in many deaths, as eating disorders are reported to have one of the highest, if not the highest, mortality rate among all mental disorders [1–4].

Generally, ED involve heightened preoccupation with food, such as restriction, or consumption of unusually large amounts of food, which in some cases is followed by compensatory behaviours such as vomiting, use of laxatives, or overexercising [5]. In extreme cases,

model parameters are available at: https://github.com/onyskj/mbl_edbi_manuscript_v2.

**Funding:** The funding for the online participants participation was provided by the Institute for Adaptive and Neural Computation, School of Informatics, University of Edinburgh and was not covered by a specific grant. The funders had no role in study design, data collection and analysis, decision to publish, or preparation of the manuscript.

**Competing interests:** The authors have declared that no competing interests exist.

eating disorders result in death either due to severe malnutrition, morbid obesity or suicide [4, 6]. Moreover, the recovery from ED can be as low as 24% even after 10 years [7], which in combination with high mortality and a significant decrease in quality of life, calls for extended research into the roots and treatments of ED.

In addition to eating related behaviours, one of the core symptoms of ED is body image disturbance, which can be understood as a negative misrepresentation of one's body, body image preoccupation, usually involving disgust, shame and dissatisfaction [8]. We here aim to investigate how body image dissatisfaction in eating disorders affects basic mechanisms of decision making, namely, habitual (repeating of the same action as a response to a stimulus) and goal-directed (intentional and deliberate decision-making) behaviour [9]. On top of logistic regression analysis of the behavioural data [10], the mechanism will be captured in a computational, reinforcement learning (RL) framework.

## Background

### Eating disorders

There are three main subtypes of eating disorders–Anorexia Nervosa (AN), Bulimia Nervosa (BN) and Binge-Eating Disorder (BED) as in 5[th] edition of The Diagnostic and Statistical Manual of Mental Disorders [5]. All share low life satisfaction, greatly impaired quality of life, and increased mortality or suicide risk, and sometimes body image issues. A brief distinction between the types, but nowhere close to accurately describing the nuance and lived-experience of each subtype, would characterise: i) AN by extreme restriction of food and pursuit of weight loss; ii) BN by undergoing recurrent binge eating episode followed by compensatory behaviours; and iii) BED by the recurrence of uncontrollable binges, but usually without compensatory behaviours [5]. In addition, there exists an array of atypical eating disorders that do not fit the description of the main (clinical) types [11], but nevertheless constitutes a majority of cases [12]. In this paper, we do not focus on any particular type, but rather aim to explore and quantify the effect of body image dissatisfaction on decision making in broadly understood eating disorders (see Methods).

### Computational psychiatry

One possible avenue to understand and help devise treatments for eating disorders, alongside the traditional approach of psychotherapy, psychiatry and neuroimaging [13–15] is the emerging field of computational psychiatry (CP).This framework is based in the assumption that the brain's characteristic function is one of computation and information processing. As such, it offers an understanding of mental illness whereby the differences and/or errors within these computations may result in maladaptive behaviours and mental states [16]. In an attempt to describe these mechanisms, some researchers have focused on multiple modes of decision making that can be gauged with a decision-making task and quantified with a computational model that captures individual and group differences [17, 18]. Mental disorders such as schizophrenia, anxiety, and obsessive-compulsive disorder (OCD) have been linked to various significant changes in decision-making processes [19–21]. Moreover, employing computational methods to understand mental disorders promises bridging the neurobiological and psychiatric levels of descriptions of populations of interest, for example the role of dopaminergic signals in reward learning and it implications in addiction [22]. Such a link can potentially inform better and more advanced theories of mental disorders, as well as inspire new treatment approaches, and promises improved early detection leading to prevention [23].

## Computational psychiatry of eating disorders

As mentioned above, depression, OCD, anxiety, and schizophrenia have received a lot attention in the field of CP, with very promising results and theories. Eating disorders have received less attention, with only eleven papers in the last nine years [24–34]. To illustrate the difference a Google Scholar search for eating- disorder related papers:

> ("computational psychiatry" "OR" "computational" "OR" "reinforcement learning" "OR" "reinforcement" "OR" "bayesian" "OR" "decision making" "OR" "decision-making") "AND" ("anorexia" "OR" "anorexia nervosa" "OR" "bulimia nervosa" "OR" "binge" "OR" "binge eating" OR" "eating" "OR" "bulimia") [35],

returns 11 results, while a search for schizophrenia related papers:

> ("computational psychiatry" "OR" "computational" "OR" "reinforcement learning" "OR" "reinforcement" "OR" "bayesian" "OR" "decision making" "OR" "decision-making") "AND" ("schizophrenia") [36],

returns 466 results.

The main focus has been on two aspects of decision making in eating disorders. One is related to how sensitive to punishments individuals with AN are, that is how well they learn from negative feedback from the environment (in this case a decision-making task). Unfortunately, the results prove to be quite contradictory [24, 33].

The second aspect of ED that has been of interest to computational psychiatrists is the trade-off between goal-directed and habitual system utilisation in decision-making. As mentioned in the introduction, goal-directed decision-making is related to acting in the environment with a goal in mind, intentionally and deliberately. It is usually characterised by forming an internal model of the environment that describes which states and actions will bring about the best results over prolonged time [9], an idea originally put forward by Tolman [37] who considered animals forming a 'field map of the environment" while in a maze, later reframed as a 'cognitive map' [38]. On the other hand, habitual decision-making is associated with responding to stimuli in the environment in an automatic manner, usually repeating those actions that immediately yield the best results. In this case, an agent does not create a model of the environment. As such, goal-directed behaviour relies on model-based learning, which allows building an accurate model of the states and actions and their associated values that takes into account a hidden probabilistic structure of the environment. However, this kind of learning is computationally more demanding, using more resources to support the process. The habitual behaviour employs model-free learning, which updates a running score of possible states and actions, based on previous experience, without registering any hidden structures in the environment. Such a process is computationally efficient as it relies mainly on the memory of the last events [9]. To make the best of accuracy and computational efficiency, the two systems may in fact interact with each other, rather than compete for cognitive control, where the model-based system trains the model-free system by simulating the environment. Such a simulation produces transitions and rewards from which the model-free system can then learn [39].

The two systems can be captured using a well-established two-step decision-making task based on a reinforcement learning paradigm, which shows that healthy participants employ both model-free and model-based learning that trade off against each other [10]. Several studies attempted to see how this trade-off is different within eating disorders. The results are

converging and suggest that reduced model based learning is associated with high eating disorder questionnaire scores in a general population [30]. The reduction is also evident in clinical AN and BED groups [27, 34]. Moreover, a recent study by Foerde et al. [27] suggests even further reduced model-based learning contribution in AN when the task is strictly food related as compared to a monetary task. This is of particular significance for two reasons. Firstly, AN is characterised by extreme pursuit of weight-loss, which could intuitively be understood as extreme goal-directed behaviour. As such, one would expect to see increased model-based learning in AN, yet the results suggest otherwise–such "goal-pursuit" of weight-loss in AN is actually habitual, almost compulsive. Secondly, the study includes an additional food-related task, which aims to test whether the goal-directed deficits observed during the monetary task were due to monetary rewards not being perceived as motivating enough for the AN participants to employ model-based strategy (domain-specific). The results from this study show that such deficits are in fact domain-general, where model-based learning is even further reduced during the food-related task. Another study of potential interest is that on the effect of hunger on the two systems by Van Swieten et al. [40]. The authors provide evidence that hunger improves healthy participants' overall performance in the two-step decision task by increasing reliance on model-free control, without affecting model-based learning.

## Problem statement, objectives and hypothesis

Evidently, there is a need for more computational psychiatry research about eating disorders, given the high mortality. A particular aspect that has not been previously investigated is that of body image disturbance.

As a first step in this direction, we aim to explore the effect of body image dissatisfaction on decision making in a population marked by negative appearance beliefs and disordered eating by implementing a two-step decision-making task that captures both model-based and model-free contribution [10]. The task is given to a population that scores high on an eating disorder, and body image disturbance questionnaires. For comparison, a healthy control group (HC) is also recruited. We employ two task conditions: first, a standard monetary version [10], with a changed narrative to achieve higher engagement; second, a body-image related version, which aims to target dissatisfaction with body image and its manifestation in decision making. The introduction of a body image dissatisfaction condition aims to create a context similar to that in which ED finds themselves during everyday struggle—excessive worry about body image and how to change it through dieting. Secondly, our design intends to test whether the goal-directed deficit may extend beyond the domain of purely monetary tasks, into that more specific to negative appearance beliefs. This partly mirrors Foerde et al. [27], but unlike the authors' objective ours may not necessarily capture whether the said deficits are domain-general.

We hypothesise that: 1) In the monetary (neutral) condition and as described by previous studies, the group with ED will report significantly decreased model-based learning as compared to HC, and no difference in model-free learning as previously shown [27, 30]; 2) Model-based learning will be further decreased in the body image dissatisfaction condition in the group with ED. HC will not report any significant difference between conditions.

## Materials and methods

### Participants

An online study was performed. Participants for the study were recruited using Prolific, which is an online participant recruitment service used mainly for research and academic purposes

[41]. The platform provides a pool of participants that are reliable and allows to custom screen for different groups before recruitment.

Since two groups were needed–a healthy control (HC) and an eating disorder (ED) group—we applied two separate sets of pre-screening criteria on Prolific (pre-set filters that are available in the study designer). For HC we looked for people who:

a. "Have never gone on a diet in the past."

b. "Do no currently for at least one week restrict food intake to manage weight."

c. "Have no diagnosed mental health condition that is uncontrolled (by medication or intervention) and which has a significant impact on your daily life/activities."

d. "Do not have or have not had a diagnosed, on-going mental health illness or condition."

For ED, participants had to select that they have gone on a diet, as in criterion (a), and that they restrict food intake to either lose or maintain weight, as in criterion (b). Moreover, to ensure data of satisfactory quality both groups had additional criteria to meet:

i. age between 18–38

ii. have normal or corrected-to-normal vision

iii. female as an assigned sex at birth

iv. an approval rate of 98%

v. a minimum of 20 previous submissions on Prolific.

Participants first completed a range of questionnaires (sub-study 1) to be then further selected to complete a decision-making task (sub-study 2). The questionnaires were hosted on Qualtrics (an online survey software) [42], while the decision-making task was hosted partly on Qualtrics and partly on Pavlovia (an online behavioural experiment platform) [43], designed using PsychoPy3 v. 2020.1.2 software [44]. For each sub-study, participants were paid at a rate of £6.25 per hour. The questionnaire took on average 7 minutes to complete, while the task took on average 27 minutes. Subjects were based all around the world. The study was approved according to the University of Edinburgh's Informatics Research Ethics Process, with an RT number 2019/48215. After being presented with a participant information sheet, participants gave explicit consent to take part in the study using an online consent form hosted on Qualtrics.

After further selection based on questionnaires and application of exclusion criteria for the decision-making task (see S1 File), 35 (ED group) and 32 (HC group) participants were selected for data analysis.

## Power analysis

A prior power analysis was performed to determine the sample size needed to detect a significant difference (at the 5% level) between groups in the neutral condition. Following a similarly designed study by Foerde et al. [27], for calculation 1, we aimed for a medium effect size of d = 0.75 (two-tailed t-test between two independent means), and the power $(1-\beta)$ = 0.8. The sample size calculation using G*Power software [45] suggested recruiting 29 participants per group. Moreover, we calculated (calculation 2) the sample size needed to detect the difference between conditions in the ED group (two-tailed t-test between two dependent means). Setting the effect size at d = 0.75 and the power at $(1-\beta)$ = 0.8, resulted in n = 16 for the ED group.

Additional post hoc power analysis was performed as above. Using the results from this study with 35 participants in the ED group and 32 in the HC group, calculation 1 revealed effect size of d = 0.88 and power $(1-\beta)$ = 0.94, while calculation 2 revealed effect size of d = 1.04 and power $(1-\beta)$ = 0.99.

## Self-report questionnaires

In the first part of the study, participants answered questions about their age, gender, weight, and height. They also completed three questionnaires—Eating Attitudes Test (EAT-26) [46], Appearance Anxiety Inventory (AAI) [47] and The Obsessive–Compulsive Inventory (OCI-R) [48]–to assess the spectrum of: an eating disorder, body image disturbance, and obsessive-compulsive behaviours, respectively.

In order to divide participants into HC and ED groups, cut-off points were applied for the questionnaires as in the Table 1. For EAT-26, it is suggested to use a cut-off of 11 for subclinical populations [49] who display disordered eating behaviours that could warrant further clinical diagnosis, while a score of 20 and above is considered to be within a clinical range [46]. For AAI, a few cut-off points have been suggested: a score above 6 may suggest some issues with body image [50], while scores between 15–20 is reported in an appearance-concerned adult population, and a score above 20 indicates a high-risk group for body image disturbance [51]. Lastly, an additional criterion for HC group was applied so that they do not display worrying obsessive-compulsive behaviours as these have also been associated with reduced model-based learning [30]. This can be measured with OCI-R, where an optimal cut-off of 21 was suggested [48]. Given such varied recommendations, we decided to follow an approach based on sensitivity and specificity measures, where high sensitivity should ensure that participants are very unlikely to be diagnosed with a condition, while high specificity provides more confidence that participants would be diagnosed appropriately [52]. Hence, for EAT-26 [49] and AAI [53] we set a threshold of ≤10 for HC and ≥14 for ED; for OCI-R [48] we set a cut-off for HC at ≤10. The achieved specificity and sensitivity measures for each cut-off are given in Table 1.

## Two-step decision making task

Selected participants were asked to complete an established two-step decision-making task shown to quantify model-based and model-free learning contribution [10]. Before proceeding with the task, participants were asked to select a body type that is most similar to their own from a diverse range of body types in S1 Fig. Furthermore, they were instructed on how to complete the task by reading through an illustrated tutorial. After reading the instructions, they completed 25 trials of the task as a practice.

**Table 1. Cut-off points for ED and HC on EAT-26, AAI, and OCI-R questionnaires.**

| | Group | | | | |
|---|---|---|---|---|---|
| **Measure** | **HC** | | | **ED** | |
| | cut-off | sensitivity | | cut-off | specificity |
| EAT-26 | ≤10 | 70.6% | | ≥14 | 70.5% |
| AAI | ≤10 | 95% | | ≥14 | 79% |
| OCI-R | ≤10 | 92.1% | | any | - |

For each cut-off a sensitivity measure is provided for HC a specificity measure for ED.

Lastly, two attention checks were implemented in the questionnaires to filter out participants not taking part in the study in good faith.

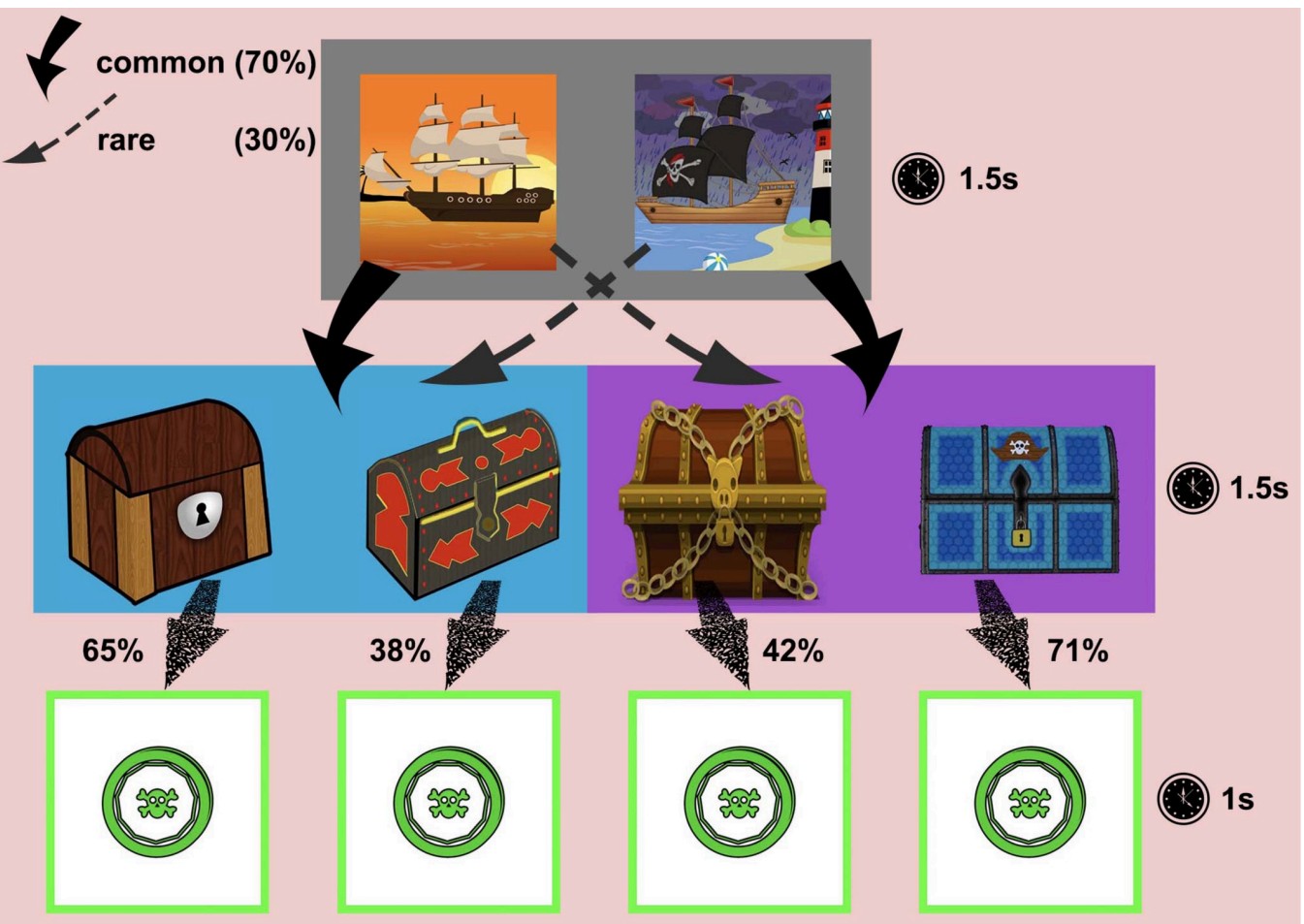

**Fig 1. The structure of the two-step decision-making task.** In the first stage (grey), a dashed arrow corresponds to a rare transition with the probability of 30%, while the bold arrow corresponds to a common transition with the probability of 70%. In the second stage, the speckled arrow is associated with the probability of receiving a reward. The duration of each stage is noted to the right (stage one, stage two, reward).

The task reproduces that of Daw et al. [10] but was conceived as a treasure game to make it as engaging and easy to understand as possible (Fig 1). In the first stage (grey stage), participants are presented with a choice between two stimuli—on the left, a ship against a sunset, while on the right, a pirate ship near a lighthouse. They are asked to play a role of a treasure hunter and decide which ship to board. Each ship can sail to two destinations—a blue island and a purple island. The game has a hidden structure, which participants tend to learn with time, such that the normal ship sails to the blue island (left) 70% of the time (common transition) and to the purple island (right) 30% of the time (rare transition), while the pirate ship sails to the blue island 30% of the time and to the purple island 70% of the time.

Once the participant boards the chosen ship, they are taken to one of the islands (stage two), where two chests await them. Here, they are asked to quickly select the chest to try their chances at finding a treasure (a pirate coin). However, each chest is assigned a probability of containing the coin, which evolves over time according to a Gaussian random walk with a standard deviation $\sigma = 0.0275$, as in the Fig 2 (more details in the S1 File). One of the strategies is to track, over time, which chest is the most favourable, i.e. has the highest chance of yielding the coin. After the chest is opened and the participant receives the coin (or not), they are taken back to the first stage to repeat the trial.

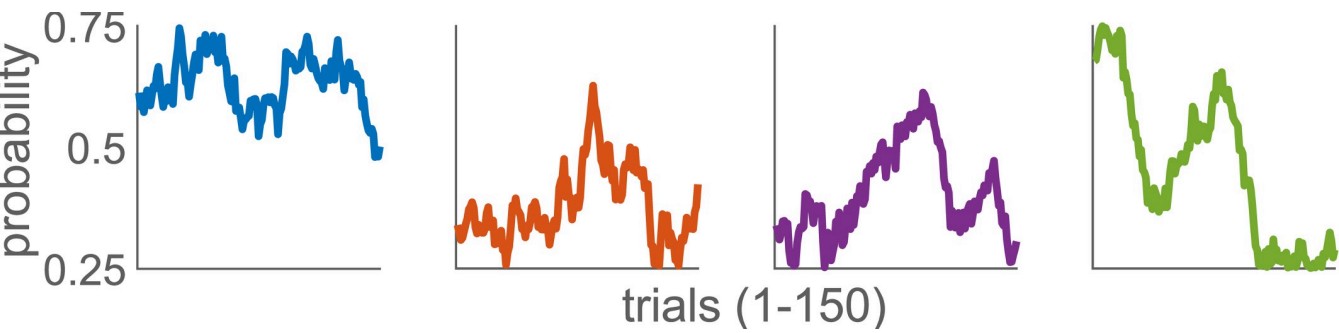

**Fig 2. The sample evolution of the reward probability for each of the chest in stage 2.** The probabilities (in the order as in Fig 1) evolve over 150 trials, according to a Gaussian random walk with σ = 0.0275. One set of two evolutions always starts randomly in a range [0.58, 0.72], while the other set of two starts in [0.31, 0.45]. The values of probabilities are bounded in [0.25, 0.75].

There are two conditions, 150 trials each. Each condition consists of two series of 75 trials, with a short break in between to counteract a possible loss of attention or tiredness. The neutral (NT) condition is exactly like in Fig 1, with a pirate coin as a reward (Fig 3A), and, aside from the narrative change, follows exactly from Daw et al. [10] and Gillan et al. [30]. The other condition, the body image dissatisfaction (BID) condition, aims to gauge the effects of body image dissatisfaction on decision-making. The BID condition is different from the neutral condition in that the reward is a pirate coin next to a body type that the participant selected as most similar to their own (Fig 3C).

For a balanced design half of the participants in each group completed the neutral condition first, followed by the BID condition. The other half completed the task in a reverse order. All results are then based on the average across two subgroups.

## Model-based and model-free learning in the task

The task allows to capture the contribution of model-based and model-free learning. Thanks to its stage-like structure and probabilistic nature, we can easily distinguish between two strategies a participant can have, usually a mix of the two.

For the purpose of this example, we focus on two extreme cases [30]. On one hand, an agent could utilise only model-free learning strategy. This means that they would make their decision as to which ship and chest to select purely based on the reward they received (or lack thereof), usually repeating the rewarded action, regardless of the transition type. On the other

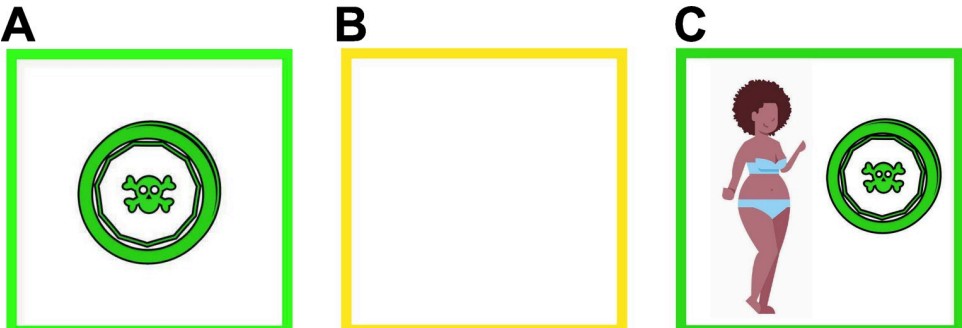

**Fig 3. Possible reward outcomes during the task.** In the neutral condition, participants could receive either (A) or (B), while in the BID condition they could receive either (C) or (B). An empty box (B) indicates no reward. The body type was selected by the participant before the task, more body type examples can be found in S1 Fig.

hand, a participant who uses only model-based strategy, takes into account the learnt transition structure of the task, on top of the knowledge about previous rewards. Moreover, they track the probability of receiving the reward to know which chest is the most favourable. In other words, they chose the same ship if the previous trial had a common transition with a reward or a rare transition and no reward.

## Mixed-effects logistic regression of raw choice data

First, a mixed-effects logistic regression analysis of raw choice data was performed to quantify the model-based and mode-free learning effects [10]. The analysis focuses only on the choices made in the first stage of the task and how these are influenced by received a reward and transition type in the previous trial. The regression models the probability of repeating the same choice, p(stay) and how it is influenced by reward (rewarded = 1, unrewarded = -1) and transition (common = 1, rare = -1), and their interaction. The main effect of reward is interpreted as model-free learning contribution and the reward x transition interaction as model-based learning contribution. These effects were regressed alongside with group, condition, and age (z-scored) as fixed effects as below:

$$\text{glmer}(\text{stay} \sim r * \text{transition} * (\text{group} * \text{condition} + \text{age\_z}) + (r * \text{transition} + 1|\text{sub}), \text{ family} \\ = \text{binomial}),$$

The estimates were obtained with the lme4 package in R, using Bound Optimization by Quadratic Approximation (bobyqa) with 1e5 functional evaluations [30].

## Stay probabilities calculation (frequency-based)

As an additional measure, for each participant we calculated the following probabilities p(stay|rewarded, common), p(stay|unrewarded, common), p(stay|rewarded, rare), p(stay|unrewarded, rare) based on the frequency of repeated choice in each of the four (reward, transition) cases. These were used for additional regression analyses and figures.

## Model-based score calculation (frequency-based)

Yet another measure that helps to capture model-based learning contribution that is independent of the reinforcement learning is the model-based score (MB score) [54]. This is calculated as follows:

$$\text{MBscore} = \text{p}(\text{stay}|\text{rewarded, common}) - \text{p}(\text{stay}|\text{rewarded, rare}) \\ - \text{p}(\text{stay}|\text{unrewarded, common}) + \text{p}(\text{stay}|\text{unrewarded, rare})$$

## Model-based and model free learning—an RL model

Finally, the full computational RL model, described and quantified below [27, 30, 55], incorporates choice data from two stages of the task, For simplicity, we begin with the update equation of the state-action value function, $Q^{II}(s_2, c_2)$ for the second stage (II) states and actions. There are two possible stage two states $s_2$: 1—the blue island; 2—the purple island, such that $s_2 \in \{1,2\}$. On each island, an agent can make two separate choices $c_2$: L—open the left chest; R—open the right chest, such that $c_2 \in \{L, R\}$. Moreover, after opening the chest, the agent can receive the reward, $r \in \{0,1\}$, where 0 corresponds to an empty chest, and 1 corresponds to the pirate coin. At the start of the task all value functions are initialised at 0.5. On any trial, $t$, we update the value function $Q^{II}(s_2, c_2)$ of the visited state and action taken as in the Eq 1. The subscript, $t$, indicates the current trial values, whereas $t+1$ indicates the values at the following trial. $\alpha$ is

the learning rate.

$$Q_{t+1}^{II}(s_{2,t}, c_{2,t}) = (1 - \alpha)Q_t^{II}(s_{2,t}, c_{2,t}) + r_t \tag{1}$$

To calculate the probability of making a choice $c \in \{L, R\}$ at stage two, we use the *softmax* function as in Eq 2, with an inverse temperature parameter $\beta_2$ quantifying the influence of the value function on making the choice.

$$P(c_{2,t} = c) = \frac{\exp\{\beta_2 Q_t^{II}(s_{2,t}, c)\}}{\sum_{c' \in \{L,R\}} \exp\{\beta_2 Q_t^{II}(s_{2,t}, c')\}} \tag{2}$$

In stage one (I), we directly see how model-free and model-based learning play their part. Here, we have two sets of update equations. The first set, Eqs 3–5 is model-based, where the agent uses their knowledge about the environment—which stage two state-action is the best, as well as the structure of the task (common vs. rare transition) to update the values of the ships in stage one. In this case, the value functions for both ships are updated simultaneously, such that the value of choosing a ship is the weighted sum (by transition probability) of the maximums over stage two actions values.

$$\boldsymbol{Q_t^{MB}} = [Q_t^{MB}(c_{1,t} = L), Q_t^{MB}(c_{1,t} = R)] \tag{3}$$

$$Q_t^{MB}(c_{1,t} = L) = 0.7 \times \max_{c_2}\{Q_t^{II}(s_2 = 1, c_2)\} + 0.3 \times \max_{c_2}\{Q_t^{II}(s_2 = 2, c_2)\} \tag{4}$$

$$Q_t^{MB}(c_{1,t} = R) = 0.7 \times \max_{c_2}\{Q_t^{II}(s_2 = 2, c_2)\} + 0.3 \times \max_{c_2}\{Q_t^{II}(s_2 = 1, c_2)\} \tag{5}$$

The second, model-free update in Eq 6, updates the values of the ships based only on the reward received at the end of the trial.

$$Q_{t+1}^{MF}(c_{1,t}) = (1 - \alpha)Q_t^{MF}(c_{1,t}) + r_t \tag{6}$$

Model-free and model-based contributions are joined together in a weighted value function over two ship choices, $Q^I$, as in Eq 7.

$$Q_t^I(c) = \beta_{MB}Q_t^{MB}(c) + \beta_{MF}Q_t^{MF}(c) + \rho I(c = c_{1,t-1}) \tag{7}$$

The contribution of each system is captured with $\beta_{MB}$ (model-based) and $\beta_{MF}$ (model-free) parameters. An additional indicator $I(c = c_{1,t-1})$ tells if the choice made on the current trial is repeated as in the previous one, with a parameter $\rho$ describing how much switching or staying is done regardless of the feedback. Finally, the probability of choosing either ship is calculated as in Eq 8, analogously to the second stage.

$$P(c_{1,t} = c) = \frac{\exp\{Q_t^I(c)\}}{\sum_{c' \in \{L,R\}} \exp\{Q_t^I(c')\}} \tag{8}$$

The model has a total of five parameters: $\beta_{MB}, \beta_{MF}, \beta_2, \alpha, \rho$.

## Model fitting

Model parameters for each group and condition were estimated using hierarchical Bayesian approaches, which provides the best test-retest reliability for this particular model, and follows the procedure as in Brown et al. [56]. The estimation was performed using RStan package (v.2.21.0) [57] in R (v. 4.0.2) based on Markov Chain Monte Carlo techniques (No-U-Turn Hamiltonian Monte Carlo). Each parameter was estimated with a mean, scale, and individual

error estimates. The learning rate $\alpha$ was constrained to (0,1) with an inverse logit transformation, while the means of $\beta_{MB}$, $\beta_{MF}$, were bounded below by 0. We used weakly informative and uninformative priors for the means: $\alpha{\sim}N(0,2.5)$, $\beta_{MB}{\sim}N(0,100)$, $\beta_{MF}{\sim}N(0,100)$, $\beta_2{\sim}N(0,100)$, $\rho{\sim}N(0,100)$. For the scales of each parameter, we used Cauchy(0,2.5) distribution, all constrained to be greater than 0. Individual error terms were all $N(0,1)$, where for $\beta_{MB}$, $\beta_{MF}$ these were constrained to be greater than 0. For each of the four chains we ran 2000 samples (after discarding 2000 warm-up ones). Chains were inspected for convergence and their $\hat{R}$ values were all around 1 (below 1.1). The mean value (across chains) of each parameter for each participant were used in the analysis.

## Mixed-effects linear regression of parameter estimates

The estimates for each parameter were analysed using mixed-effects linear regression to compare them between groups and conditions, with group, condition, and age (z-scored) as fixed effects per subject:

$$\text{lmer}(\text{parameter} \sim \text{group} * \text{condition} + \text{age\_z} + (1|\text{sub}))$$

## Parameter recovery

To quantify the reliability of the parameter estimates, parameter recovery was performed after data collection (see S1 File). For the parameters range compatible with the collected data, the model and fitting procedure described above provide fair to excellent reliability, with average (across parameters, groups and conditions) parameter recovery Pearson correlation coefficient (PCC), r = 0.834.

# Results

## Demographic and questionnaires summary

The screening procedures resulted in recruiting two significantly different groups—healthy control (HC) and eating disorder (ED). The summarised information, along with two-sample t-tests, can be found in Table 2. The groups are mainly characterised by average total scores on the three questionnaires, with scores significantly different between groups across all questionnaires (p<0.001). As the groups also significantly differ in age, the variable (z-scored) was included as a covariate in the regression analysis.

## Task performance—rewards and reaction times

The performance in the task was first analysed independently of the reinforcement learning model. Average characteristics were calculated for each group (detailed results in the S2

**Table 2. Summary of demographic information and questionnaire scores in each group.**

| Measure | HC (n = 32) Mean (SD) | ED (n = 35) Mean (SD) | t value | p value |
|---|---|---|---|---|
| EAT-26 | 2.94 (2.72) | 25.83 (10.67) | -11.78 | **<0.001**\* |
| AAI | 4.59 (2.87) | 23.94 (7.26) | -14.10 | **<0.001**\* |
| OCI-R | 5.5 (3.59) | 22.83 (11.09) | -8.44 | **<0.001**\* |
| BMI (kg/m$^2$) | 21.83 (4.27) | 26.07 (6.21) | -3.01 | **0.004**\* |
| Age | 26.38 (4.61) | 30.57 (4.45) | -3.79 | **<0.001**\* |

This includes means and standard deviations (SD) of EAT-26, AAI, OCI-R scores, age, and BMI, as well as t- and p- values of the two-sample t-tests.

\*—significance at the 5%-level.

Table in S1 File). These measures include: total reward in the neutral and BID condition, after completing the full task, as well as mean reaction times (RT) during the neutral, BID, and across both conditions. There were no differences in the above measures between groups.

## Collected raw choice data analysis

**Mixed-effects logistic regression analysis–collected data.** To assess the model-based and model-free contribution during the task, raw choice data was analysed with mixed-effects logistic regression of the probability of repeating the same choice in stage one (Table 3). HC group showed a significant contribution of both model-free system as indicated by the reward effect (p-value<0.001), and model-based system as indicated by the reward x transition interaction effect (p-value = 0.018). Our hypothesis predicted a significantly lower estimate for the reward × transition × groupED effect to detect a difference between groups in the neutral condition; as well as a significantly lower estimate for the reward × transition × groupED × conditionBID effect to detect a difference between condition in the ED group. However, the analysis revealed no differences between groups or conditions in terms of model-based and model-free learning. The estimated probabilities (frequency-based) for each case in each group and condition are depicted in Fig 4.

We further verified whether model-based and model-free learning strategies are used in our sample, regardless of group and condition. As such, these fixed effects were removed from the regression analysis (Table 4). The simpler model revealed significant contribution of both learning systems in the joint population (Reward effect p-value<0.001, Reward x Transition effect p-value = 0.008). Moreover, there was a significant effect of transition (p-value <0.001).

**Table 3. Random effects logistic regression for probability of staying (collected data).**

| Effects | Estimate | SE | z value | p value |
|---|---|---|---|---|
| Intercept (HC, conditionNT) | 1.68 | 0.24 | 6.86 | **<0.001**\* |
| reward | 0.92 | 0.33 | 2.77 | **0.006**\* |
| transition | -0.57 | 0.16 | -3.68 | **<0.001**\* |
| groupED | -0.46 | 0.35 | -1.33 | 0.183 |
| conditionBID | -0.21 | 0.13 | -1.56 | 0.120 |
| age_z | 0.18 | 0.17 | 1.07 | 0.285 |
| reward × transition | 0.84 | 0.36 | 2.37 | **0.018**\* |
| groupED × conditionBID | -0.07 | 0.18 | -0.36 | 0.718 |
| reward × groupED | -0.19 | 0.47 | -0.40 | 0.689 |
| reward × conditionBID | -0.03 | 0.22 | -0.13 | 0.900 |
| reward × age_z | -0.05 | 0.22 | -0.21 | 0.833 |
| transition × groupED | 0.26 | 0.21 | 1.23 | 0.219 |
| transition × conditionBID | 0.09 | 0.16 | 0.57 | 0.569 |
| transition × age_z | -0.05 | 0.09 | -0.57 | 0.566 |
| reward × groupED × conditionBID | 0.21 | 0.30 | 0.72 | 0.473 |
| transition × groupED × conditionBID | 0.11 | 0.22 | 0.51 | 0.612 |
| reward × transition × groupED | -0.43 | 0.50 | -0.87 | 0.387 |
| reward × transition × conditionBID | 0.05 | 0.27 | 0.21 | 0.837 |
| reward × transition × age_z | 0.00 | 0.23 | 0.00 | 0.999 |
| reward × transition × groupED × conditionBID | -0.30 | 0.36 | -0.85 | 0.397 |

Group, condition, age (z-scored) are treated as fixed-effect covariates per subject.

\*—significance at the 5%-level.

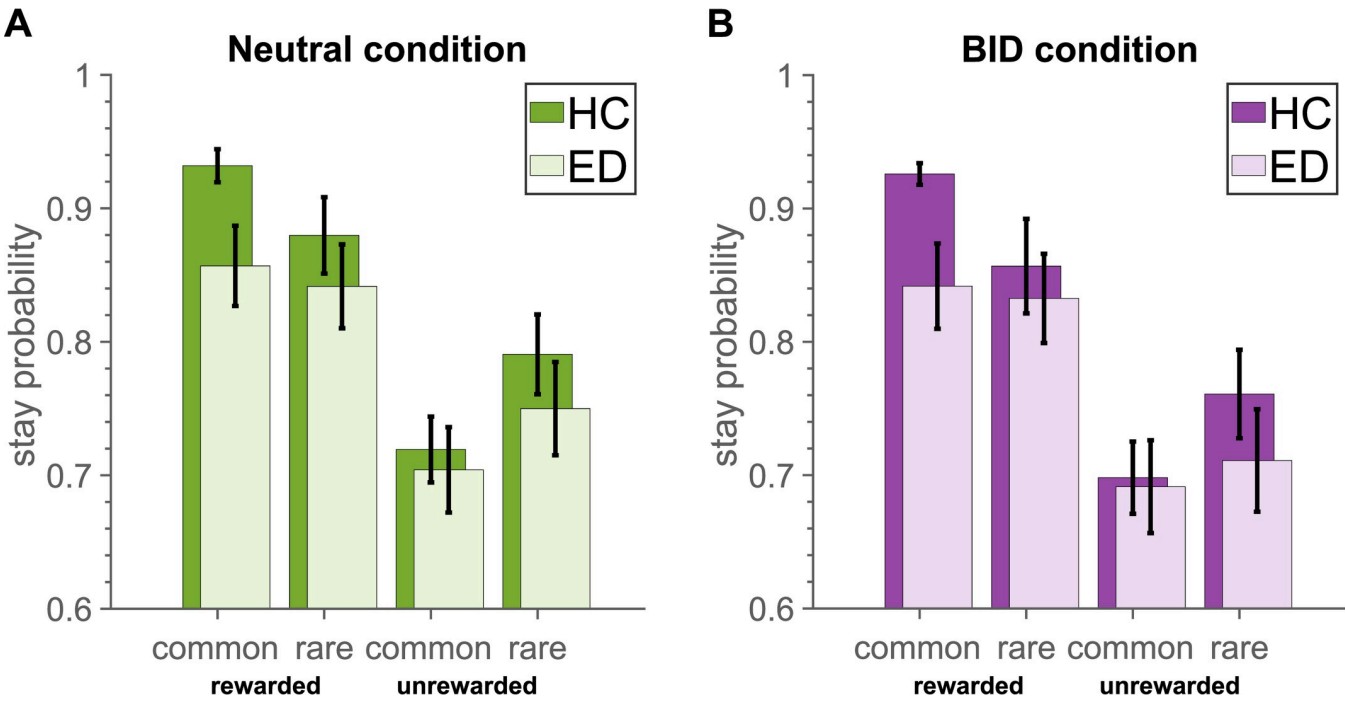

**Fig 4. Stay probabilities in the collected data.** (A) Neutral condition. (B) BID condition.

### Frequency based-probability regression–collected data

Furthermore, we estimated the probabilities of sticking to the same stage 1 choice after a common transition on the previously rewarded trial using the frequency calculation and used them in mixed-effects linear regression analysis (Table 5). This simpler comparison revealed reduced probability of staying after common, rewarded trials between groups in the neutral condition, which was just short of significance (p-value = 0.054), indicating a potentially weaker model-based learning capacity in ED. There were no differences between conditions in either group.

**Table 4. Random effects logistic regression for probability of staying (collected data).**

| Effects | Estimate | SE | z value | p value |
|---|---|---|---|---|
| Intercept | 1.31 | 0.15 | 8.45 | **<0.001**\* |
| r | 0.87 | 0.20 | 4.30 | **<0.001**\* |
| transition | -0.36 | 0.09 | -4.07 | **<0.001**\* |
| age_z | 0.08 | 0.16 | 0.48 | 0.628 |
| r × transition | 0.56 | 0.21 | 2.64 | **0.008**\* |
| r × age_z | -0.07 | 0.20 | -0.34 | 0.732 |
| transition × age_z | 0.02 | 0.09 | 0.18 | 0.857 |
| r × transition:age_z | -0.13 | 0.21 | -0.60 | 0.552 |

Age (z-scored) is treated as fixed-effect covariates per subject.

\*—significance at the 5%-level.

**Table 5. Random effects for probability of staying after a common transition and a rewarded trial based on collected data (frequency estimate).**

| Effects | Estimate | SE | t value | p value |
|---|---|---|---|---|
| Intercept (HC, conditionNT) | 0.93 | 0.03 | 35.79 | **<0.001**\* |
| groupED | -0.07 | 0.04 | -1.96 | 0.054 |
| conditionBID | -0.01 | 0.01 | -0.46 | 0.649 |
| age_z | 0.00 | 0.02 | -0.09 | 0.930 |
| groupED:conditionBID | -0.01 | 0.02 | -0.50 | 0.622 |

Age is treated as fixed-effect covariates per subject.

\*—significance at the 5%-level.

### MB score analysis–collected raw choice data

Lastly, we calculated MB scores in the collected data (Fig 5). These were regressed against group and condition variables (Table 6). There were no differences in MB scores between groups and conditions.

### RL parameter estimates—model-based and model-free learning

Next, we performed a more sensitive analysis, using a RL model, which, unlike the previous analyses which are based on trial averages, takes into account incremental learning over many trials. As it is not uncommon for the results from model-agnostic analysis to differ from computational modelling results [58–60], we fit parameters for the reinforcement learning model as described in the Methods. A comparison of average parameter estimates with standard errors (SE) of $\beta_{MB}$, $\beta_{MF}$ between groups and conditions can be found in Fig 6.

The results from the mixed effects linear regression model for each parameter, showing a significant difference between groups and conditions can be found in Tables 7–10.

As hypothesised, the model-based contribution (as quantified with $\beta_{MB}$ parameter) in the neutral condition is decreased in ED as compared to HC (Table 7; p-value = 0.016). Moreover, there is a further reduction in model-based learning in the BID condition in ED (p-value = 0.003), which is not present in the HC group (p-value = 0.060).

Surprisingly, model-free learning is significantly reduced in the neutral condition in the ED group (compared to HC p-value <0.001, Table 8). Furthermore, model-free learning is slightly attenuated in the BID condition in HC (p-values = 0.011).

In addition to model-based and model-free learning, there was an increase in the learning rate between conditions for HC (p-value = 0.004; Table 9), as well as a reduction in the $\beta_2$ inverse temperature parameter estimate in the ED group in the neutral condition (p-value = 0.001; Table 10), potentially indicating a more exploratory choice strategy in the second stage.

### Correlations of $\Delta\beta_{MB}$ with other covariates

To check how the difference in model-based learning between condition correlates with questionnaire scores and demographics (age, EAT-26, AAI, and OCI-R scores all z-scored), we introduced $\Delta\beta_{MB}$ that captures the difference in $\beta_{MB}$ between conditions:

$$\Delta\beta_{MB} = \beta_{MB,BID} - \beta_{MB,NT}$$

when negative, this would indicate the BID condition had a reducing effect on model-based learning.

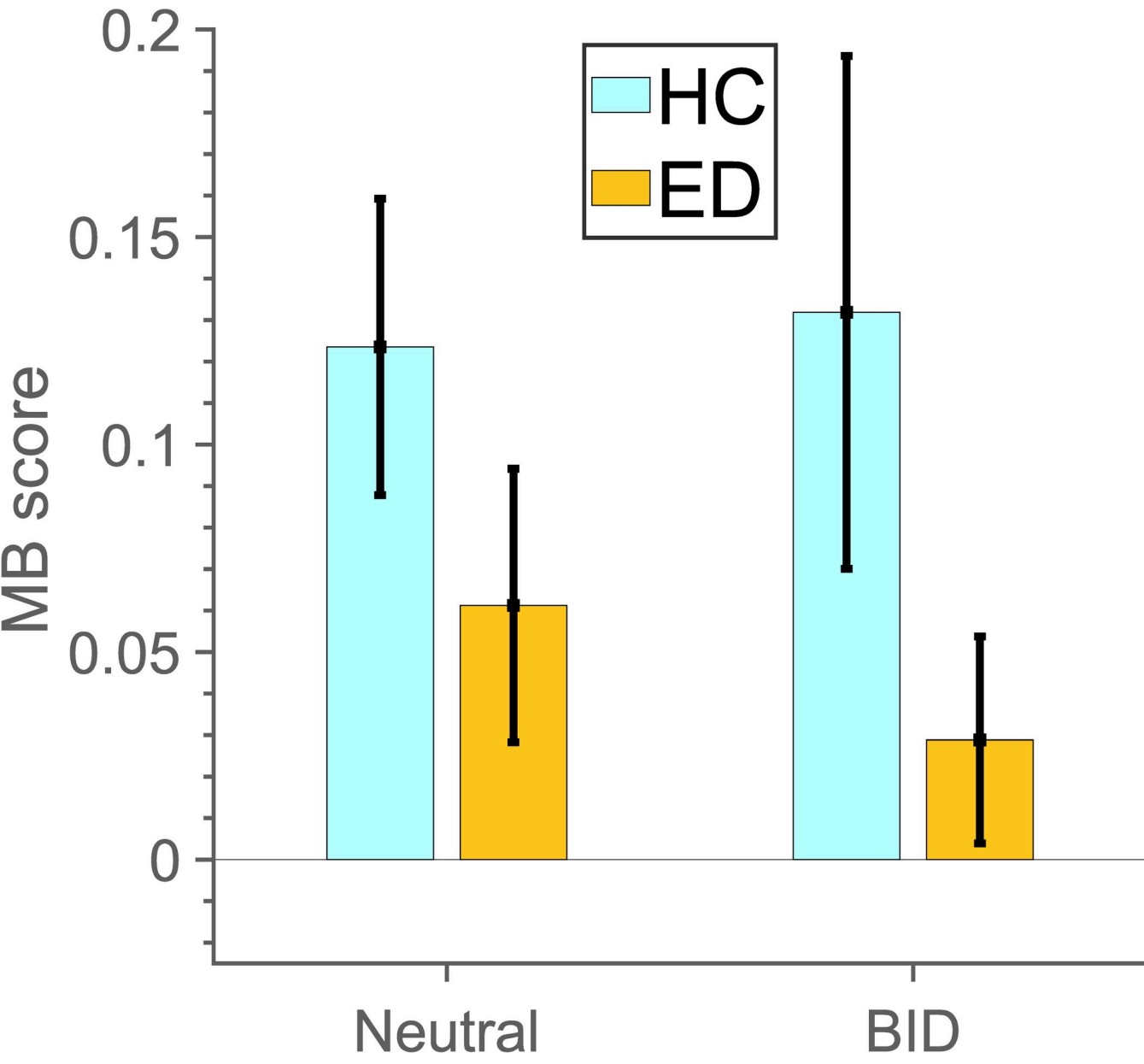

**Fig 5. MB scores per group and condition in collected data.**

The correlation plots with Pearson correlation coefficient (PCC), *r*, and p-values for the hypothesis of no relationship between $\Delta\beta_{MB}$ and the covariates can be found in Fig 7. We found significant negative correlations with EAT-26 and AAI scores and OCI-R scores as in Table 11 (*r* = -0.312, r = -0.314, *r* = -0.316 with p-value = 0.01, p-value = 0.01, p-value = 0.009). Correlation between questionnaire scores can be found in the S7 Table in S1 File.

## Discussion

The current study focused on a strikingly missing element in computational psychiatry research on eating disorders—body image dissatisfaction. In particular, model-based learning was explored to identify whether deficits in goal-directed learning manifest alongside negative

**Table 6. Random effects linear regression for MB score (collected data).**

| Effects | Estimate | SE | t value | p value |
|---|---|---|---|---|
| Intercept (HC, conditionNT) | 0.12 | 0.04 | 2.80 | **0.006***|
| groupED | -0.06 | 0.06 | -0.93 | 0.354 |
| conditionBID | 0.01 | 0.04 | 0.20 | 0.845 |
| age_z | -0.01 | 0.03 | -0.20 | 0.843 |
| grouped × conditionBID | -0.04 | 0.06 | -0.69 | 0.490 |

Age is treated as fixed-effect covariates per subject.

*—significance at the 5%-level.

appearance beliefs, with the hypotheses of decreased model-based learning between groups (ED vs. HC) and between conditions in ED.

First of all, the findings from a two-step decision-making task show a significantly smaller contribution of model-based learning in a population characterised by high body image dissatisfaction and disordered eating as compared to the HC group. This replicates previous findings that suggest a decreased model-based learning, in a monetary condition of the two-step decision-making task, in groups characterised by disordered eating and compulsive behaviours [27, 30, 34]. We found a further model-based learning reduction in the population of interest in the condition that was meant to involve body image dissatisfaction (as implemented in the BID condition). We did not find such an effect in the healthy control. These results support the set hypotheses regarding model-based learning, suggesting that goal-directed deficits extend beyond purely monetary tasks and are more pronounced in the domain of body image preoccupation. This mirrors previously established results of further goal-direct deficits in AN when the task is food-related [27], which demonstrate that such deficits affect multiple domains of decision making. Additionally, the reduction in model-based learning between groups, and between conditions in the ED group was not associated with age.

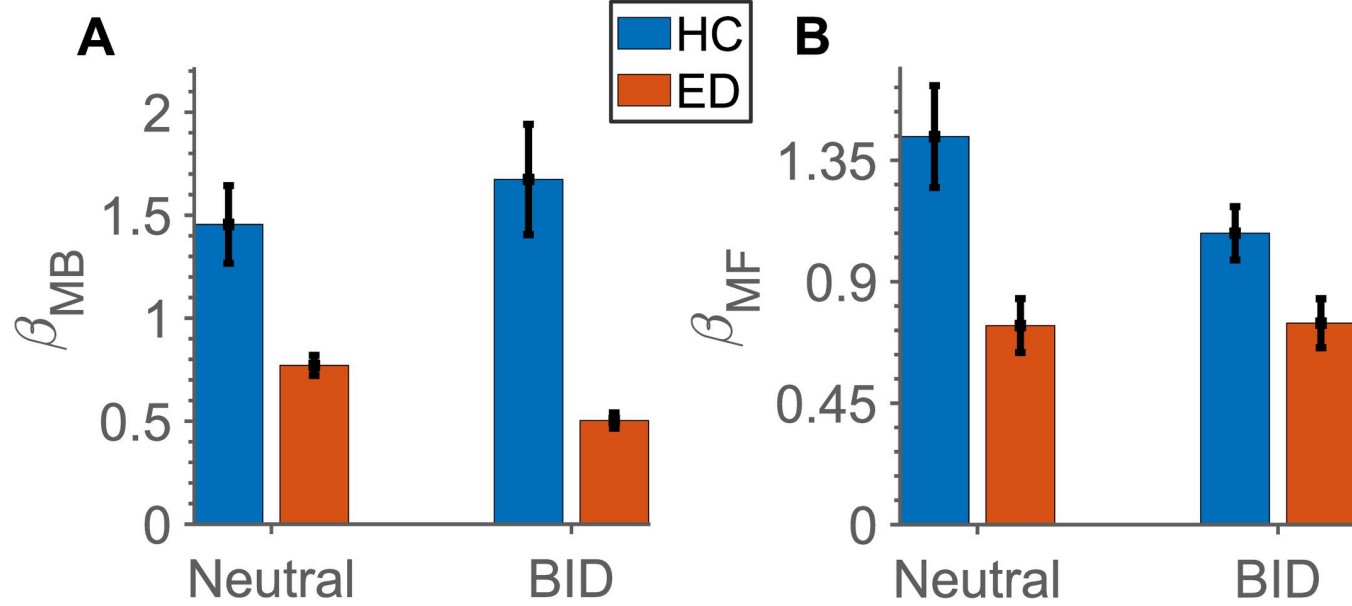

**Fig 6. Estimated model parameters.** Mean standard error (SE) of the estimated model parameters in HC (blue) and ED (red) during each condition (neutral and BID). (A) $\beta_{MB}$, (B) $\beta_{MF}$.

**Table 7. Mixed effects linear regression analysis of model-based learning parameter $\beta_{MB}$.**

| Effects | Estimate | SE | t value | p value |
|---|---|---|---|---|
| Intercept (HC, conditionNT) | 1.42 | 0.17 | 8.23 | **<0.001**\* |
| ED group | -0.61 | 0.25 | -2.46 | **0.016**\* |
| BID condition | 0.22 | 0.11 | 1.92 | 0.060 |
| Age | -0.09 | 0.12 | -0.75 | 0.454 |
| ED group × BID condition | -0.49 | 0.16 | -3.08 | **0.003**\* |

Group, condition, age (z-scored) are treated as fixed-effect covariates per subject.

\*—significance at the 5%-level.

**Table 8. Mixed effects linear regression analysis of model-free learning parameter $\beta_{MF}$.**

| Effects | Estimate | SE | t value | p value |
|---|---|---|---|---|
| Intercept (HC, conditionNT) | 1.56 | 0.14 | 10.94 | **<0.001**\* |
| ED group | -0.70 | 0.21 | -3.81 | **<0.001**\* |
| BID condition | -0.32 | 0.14 | -2.62 | **0.011**\* |
| \*Age | 0.02 | 0.09 | 0.30 | 0.764 |
| ED group × BID condition | 0.52 | 0.20 | 1.94 | 0.057 |

Group, condition, age (z-scored) are treated as fixed-effect covariates per subject.

\*—significance at the 5%-level.

**Table 9. Mixed effects linear regression analysis of learning rate parameter $\alpha$.**

| Effects | Estimate | SE | t value | p value |
|---|---|---|---|---|
| Intercept (HC, conditionNT) | 0.61 | 0.05 | 12.60 | **<0.001**\* |
| ED group | -0.09 | 0.07 | -1.33 | 0.188 |
| BID condition | 0.15 | 0.05 | 3.02 | **0.004**\* |
| Age | 0.02 | 0.03 | 0.50 | 0.619 |
| ED group × BID condition | -0.10 | 0.07 | -1.43 | 0.159 |

Group, condition, age (z-scored) are treated as fixed-effect covariates per subject.

\*—significance at the 5%-level.

**Table 10. Mixed effects linear regression analysis of inverse temperature parameter $\beta_2$.**

| Effects | Estimate | SE | t value | p value |
|---|---|---|---|---|
| Intercept (HC, conditionNT) | 1.63 | 0.12 | 13.06 | **<0.001**\* |
| ED group | -0.60 | 0.18 | -3.34 | **0.001** |
| BID condition | -0.03 | 0.11 | -0.25 | 0.802 |
| Age | 0.10 | 0.08 | 1.23 | 0.224 |
| ED group × BID condition | 0.09 | 0.16 | 0.56 | 0.576 |

Group, condition, age (z-scored) are treated as fixed-effect covariates per subject.

\*—significance at the 5%-level.

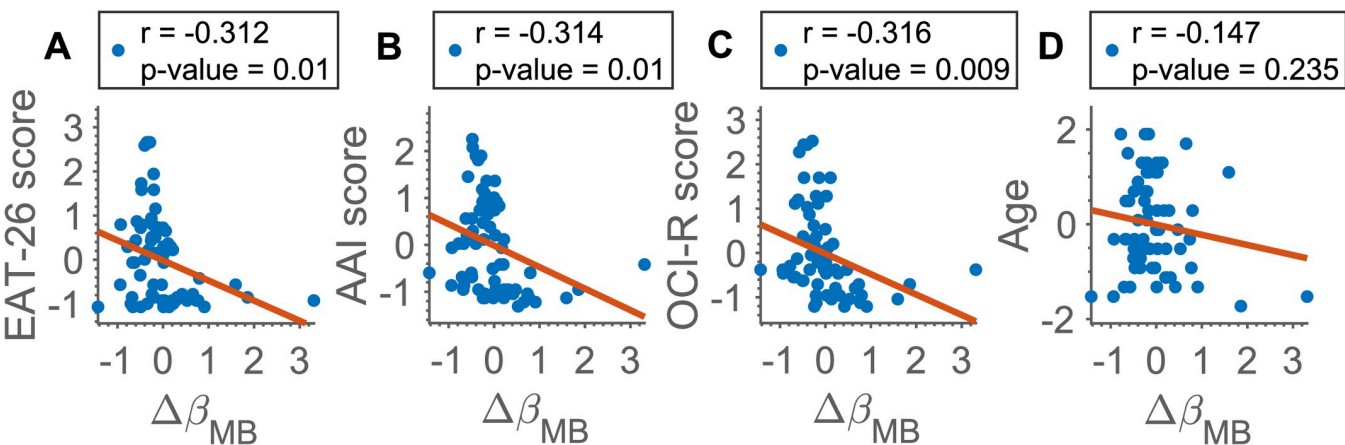

**Fig 7. Correlation of EAT-26, AAI, OCI-R and age measurements (z-scored) with $\Delta\beta_{MB}$.** Correlation is across both groups. Pearson's correlation coefficient (r) and p-values are included.

Surprisingly, contrary to one of our hypotheses, we found decreased (BID vs Neutral) model-free learning in the HC group, as well as reduced (ED vs HC) model-free learning between groups in the neutral condition. This is also contradictory with previous studies that find no difference in model-free learning between AN and HC groups in the monetary two-step task [27, 30] or increased reliance on model-free learning in a BED group [34]. One potential reason for the difference of results is the high heterogeneity in the estimates of model-free learning parameter both in ED and HC group, which may have skewed the results. In fact, HC group, in general, has more heterogeneous estimates of parameters than ED. This may be due to the HC group being in fact composed of different subgroups, possibly even along the eating disorder continuum, for example if they were not entirely sincere when answering the questionnaires. Alternatively, ED may exhibit a generally reduced learning capability (see below for further discussion), while the difference with Voon et al. [34] study may also stem from a different parameterisation of the two systems. Unlike in Voon et al. [34], our computational model does not include the parameter w that forces the two systems to trade-off against each other. In their case, the reported reduction in model-based learning necessarily results in an increase of model-free learning, while this is not necessarily the case for us.

Furthermore, we found a significant correlation of the difference between the measures of model-based learning in the neutral and BID condition with EAT-26, AAI scores and OCI-R scores, with a similar correlation coefficient. This may suggest that the model-based difference between conditions captures a similar psychiatric dimension common to all three

**Table 11. PCC, $r$, between $\Delta\beta_{MB}$ and the covariates across HC and ED groups.**

|  | Covariates | | | |
|---|---|---|---|---|
|  | **EAT-26** | **AAI** | **OCI-R** | **Age** |
| $r$ | -0.312 | -0.314 | -0.316 | -0.147 |
| p-value | **0.01**[*] | **0.01**[*] | **0.009**[*] | 0.235 |

P-values for the hypothesis of no relationship between the $\Delta\beta_{MB}$ and covariates (EAT-26, AAI, OCI-R scores and age —all z-scored) are included.

[*]—significance at the 5%-level.

questionnaires, which would also be in line with a significant correlation between EAT-26, AAI, and OCI-R scores, as well as with previously reported results [61]. As such, our study may highlight dimensionally-grounded approach to mental illness, similarly to a previous online behavioural study implicating reduced goal-directed learning in compulsive behaviour and intrusive thought dimension [30].

Lastly, the logistic regression of the raw choice data showed the two populations employ both learning systems. However, to detect the differences between groups and conditions we used a more sensitive RL model analysis that takes into account the incremental learning from many trials, as well as stage two choices, which are not part of the logistic regression model. This followed the same approach as in previous studies on goal-directed deficits in anorexia nervosa [27] and in a population marked by compulsive behaviour and intrusive thought [30].

## Implications

First of all, as far as we know, this is the first study that looked at the body image dissatisfaction from a computational perspective. The fact this phenomenon is associated with decreased contribution of model-based learning provides some support for the mechanism of extreme habitual body preoccupation. The fear of gaining weight and a goal of either losing or not gaining weight, evolves over time into a rigid and habitual body checking, which may be based in a shift away from the model-based system, in a similar fashion as in addiction [62] where, over-time, such goal becomes overvalued [63]. When an individual's body type is displayed onto a screen it may act as a trigger for the habitual behaviour of body comparison and concern. As such, a body-preoccupied state leads to significant reduction in the model-based capabilities that could be allocated towards task completion in a more goal-directed manner, as is done in the neutral condition. Our result adds to the discussion reopened by Foerde et al. [27] about domain-general/specific deficits in goal-directed learning. It provides support for the view that goal-directed control impairments are not specific to purely monetary tasks but suggests that it could be aggravated in conditions that trigger body image preoccupations.

The high heterogeneity of parameter estimates (model-based/free) in the healthy control may suggest that some healthy controls share some traits with the ED group. This might be related to traits that we have not directly explored in our study but that have been associated with deficits in model-based learning, e.g. alcohol addiction, or impulsivity [30], despite partially controlling for the effect of compulsivity on model-based learning [30] by excluding participants with high OCI-R scores in the healthy control group. Alternatively, it is possible that some participants from a healthy control group do in fact have body image preoccupation issues. A potential factor in group misclassification could be a wide-spread and widely accepted societal preoccupation with dieting, looks as well as the phenomenon of body shaming that could render the recruitment of a 'healthy' population difficult [64, 65]. This could manifest in participants as selecting the option of "no past diet experience" in the screening stage despite 'clean eating' they might engage in as a widely accepted 'health standard', which is actually emotionally distressing and linked with functional impairments [66]. As such, the heterogeneity in the parameter estimates in the healthy control may highlight a blurred boundary between health and dieting, and indeed call into question the possibility of a truly healthy control group within this field of research.

## Limitations

Group differences are here only detectable with a sensitive RL model. As such, it would be beneficial to perform a similar study on a larger online population and/or in the lab on a clinical group to strengthen the evidence for the hypothesis of this paper and test its replicability.

A possible limitation presents itself in the selection of body types/silhouettes by the participants. These may be seen as overly expressive and caricature-like, failing to accurately capture their body type, which may cast doubt on the source of the reduction in model-based learning. As such, we are currently working on a paradigm that offers a more rigorously defined range of body types for the participants to select from, where a selected image is compared to the participant's basic body parameters (e.g. weight, height), capturing the discrepancy between reality and perception [67–69]. That being said, the results of this study offer a new point for further examination: the significant change in goal-directed decision-making exclusively in the ED group, caused by the mere presence of said silhouettes.

An alternative explanation of the results may expose potential issues with the design of the BID condition. Since the silhouette is displayed with the reward, this may induce aversive behaviour (due to high-body dissatisfaction), whereby valuation of (sensitivity to) the reward is reduced. This could result in lower values of inverse temperature parameters both for model-free and model-based learning, as reward sensitivity has been shown indistinguishable from the inverse temperature [23]. Though a valid concern, we did not observe any reduction in model-free learning in the BID condition (vs. neutral) in the ED group. This suggests that the silhouettes did not have an aversive effect on reward learning, at least in the simple temporal difference learning. However, we recognise that the BID condition may have been distracting to the ED participants, due to body image preoccupation, and lead them to ruminate about their body, or to allocate more mental resources to the displayed silhouettes. As such, the BID condition could be interpreted as a modulator of the availability of cognitive resources, which has been shown to control the extent to which healthy participants employ model-based strategy [70]. As shown by Otto et al. [70], when performance of the two-step task is systematically disrupted with a high working-memory load component (a numerical stroop task [71]), participants reduce their reliance on the model-based system. An additional interpretation of the reduced inverse temperature parameters may indicate that the ED group employs a slightly different strategy where they rely more on action exploration rather than exploitation to perform well in the task [19]. To further test those possible concerns, it would be useful to verify the findings by re-running the study using a different design.

Despite employing the most reliable fitting method for this particular task and model [56], as well as applying strict exclusion criteria to participants' behavioural data and questionnaire attention checks, our parameter recovery resulted only in fair reliability of the model-based parameter. However, this is not uncommon, as studies using similar models achieve comparable measures for the model-based parameters [72, 73]. Having informally tested various models that allow for constraining of the $\beta_{MB}$, $\beta_{MF}$ parameters to be greater than 0, the only way to improve the parameter recovery would be to increase the sample size or obtain higher quality lab-based data.

Kool et al. [73] suggest that the structure of the two step decision making task used here does not accurately estimate the trade-off between model-based and model-free learning. The authors propose certain modifications to the task to increase that accuracy such as changing the drift rate in the Gaussian random walk of reward probabilities, reducing the number of stage two choices to one per state, or introducing a deterministic transition structure. However, it has been shown that in some cases of slight variations to the task structure, the reinforcement learning model will not be able to distinguish between model-based and model-free actions as efficiently as in the standard task [74], while the task in the current form has proven to yield reliable and consistent results [27, 30].

Lastly, there are a couple of steps that could be taken in order to further explore the mechanisms and effects of body image dissatisfaction on decision-making in eating disorders. A similar study could be performed on a clinical population, expanding to fMRI data collection and

analysis. This could allow to correlate the behavioural-computational changes in the ED group with neural signatures providing neurobiological basis of some of the decision-making mechanisms associated with body image dissatisfaction. For example, following Wunderlich et al. [75], it would be worth exploring whether reduced model-based learning in ED would be associated with reduced planning value representation in the anterior caudate, previously implicated in goal-directed behaviour [76]. Alternatively, focusing on correlating body-image related goal-directed deficits with the activity in the orbitofrontal cortex, could provide further support to experimental theta burst stimulation treatment approaches for eating disorders [77, 78].

## Conclusions

Given the high mortality of eating disorders, this study expands the sparse field of computational psychiatry of eating disorders that so far has focused on the general perception of reward [24, 33], and quantification of model-based learning in a neutral setting [30, 34] or in relation to food choices [27]. Since one of the prevalent aspects of eating disorders is body image preoccupation, we explored its effect on model-based learning in comparison with a healthy control. The results from the online study on a population characterised by high scores on eating disorder and body image dissatisfaction questionnaires show a significantly negative effect of body image dissatisfaction on model-based learning that is not present in the healthy control. This finding offers additional insight into the mechanisms of the disorder and the effect that the core element of the disorder, such as body image dissatisfaction, has on decision-making. Directed by the above result, further inquiry can be performed into more nuanced treatment strategies that could help to break from rigid habits and strengthen model-based capabilities of patients related to the perception of their body through well-tailored therapeutic activities.

## Supporting information

**S1 Fig.**
(EPS)

**S2 Fig.**
(EPS)

**S3 Fig.**
(EPS)

**S1 File.**
(DOCX)

## Author Contributions

**Conceptualization:** Peggy Seriès.

**Formal analysis:** Jakub Onysk.

**Investigation:** Jakub Onysk.

**Methodology:** Jakub Onysk.

**Project administration:** Peggy Seriès.

**Supervision:** Peggy Seriès.

**Visualization:** Jakub Onysk.

**Writing – original draft:** Jakub Onysk.

**Writing – review & editing:** Peggy Seriès.

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
