## [Decision Letter · Decision Letter 0]

16 May 2022

PONE-D-22-05219The effect of body image dissatisfaction on goal-directed decision making in a population marked by negative appearance beliefs and disordered eatingPLOS ONE

Dear Dr. Seriès,

Thank you for submitting your manuscript to PLOS ONE. After careful consideration, we feel that it has merit but does not fully meet PLOS ONE’s publication criteria as it currently stands. Therefore, we invite you to submit a revised version of the manuscript that addresses the points raised during the review process. I received one review from an expert in the field.  The reviewer notes many positive things about your manuscript, but also notes several issues that should be addressed before your paper is suitable for publication.  If you feel you can adequately address these issues then I invite you to submit a revised manuscript.

We look forward to receiving your revised manuscript.

Kind regards,

Darrell A. Worthy, Ph.D

Academic Editor

PLOS ONE

Journal Requirements:

"Participants payment was funded by the Institute for Adaptive and Neural Computation, Informatics, University of Edinburgh.

Reviewers' comments:

Reviewer's Responses to Questions

**Comments to the Author**

1. Is the manuscript technically sound, and do the data support the conclusions?

Reviewer #1: Yes

2. Has the statistical analysis been performed appropriately and rigorously? 

Reviewer #1: Yes

3. Have the authors made all data underlying the findings in their manuscript fully available?

Reviewer #1: Yes

4. Is the manuscript presented in an intelligible fashion and written in standard English?

Reviewer #1: Yes

5. Review Comments to the Author

Reviewer #1: I am in agreement with the authors regarding the utility of computational psychiatry, and consider their work a useful extension of current ED literature. I was pleased to see that sample size and power calculations made it into the manuscript (which remains rare), and that throughout, all statistical analyses were described thoroughly for the reader. Although none of my comments constitute major revisions, I have a few smaller points outlined below.

1. The article would benefit from a clearer exposition of how their experiment and/or results differ from those which have previously used the 2-stage task in eating disorders (refs 27 and 30). Having read these papers, it is clear how your work fits into the literature, but it would benefit the reader if you could bring this out more in the manuscript. You may also want to mention papers looking at hunger and motivation in the 2-stage task given the overlap with EDs (e.g.https://link.springer.com/article/10.3758/s13415-021-00921-w).

2. In the discussion (around line 630) you argue that the BID condition could not be impairing learning because model-free learning was unaffected. Have you considered the possibility that the body representation was specifically distracting to ED participants (who are more preoccupied with body images), resulting in mind wandering, rumination etc? If so, given that model-based learning is known to be sensitive to cognitive load, whereas model-free learning is relatively insensitive (see https://journals.sagepub.com/doi/10.1177/0956797612463080?url_ver=Z39.88-2003&rfr_id=ori:rid:crossref.org&rfr_dat=cr_pub%20%200pubmed ), is it possible that this could account for the findings you report? In other words, rather than triggering habitual behaviour (as described around line 586) could the BID condition have instead hampered model-based learning? If so this should be acknowledged in the manuscript.

3. Smaller points:

a. Line 110 – consider referencing Tolman 1948 (https://pubmed.ncbi.nlm.nih.gov/18870876/), where the idea of model-based learning originated.

b. Line 113/114 ‘without much deliberation’ this phrase is confusing. It is unclear if deliberation refers to conscious or unconscious processes. It is not clear to me that there is any deliberation in model-free decision making as the action with the highest value is simply selected from representations in the striatum.

c. Line 115/7 – it has been demonstrated by several papers (e.g. https://content.apa.org/record/2012-32653-001) that model-free learning can still be influenced by model-based information. The manuscript should reflect this as model-free valuations can take into account the probabilistic nature of the environment, albeit without an internal model.

d. Line 120 – model-free learning is influenced by more than just the previous outcome (‘last experience’). It may be influenced by hundreds of outcomes. Consider instead ‘based on previous experience’. Similarly on line 122 – previous events.

e. Consider adding a line about the implications of your work in the abstract.

4. Throughout the manuscript there are several small typos and occasional sentences with clunky grammar. Some of the errors are listed below, but it would be worth proofing the entire manuscript again to correct any unfortunate niggles:

a. Line 38 – removal of can – they do result in many deaths.

b. Line 86 – grammar – such links or such a link.

c. Line 92 – 10 papers is not ‘a couple’ it is 10.

d. Line 468 – formatting error – remove line break in mid-sentence.

e. Line 656 – on rather than o.

6. PLOS authors have the option to publish the peer review history of their article (what does this mean?). If published, this will include your full peer review and any attached files.

Reviewer #1: No

---

## [Author Response · Author response to Decision Letter 0]

7 Jul 2022

Please see our Reply to Reviewers document attached.

---

## [Decision Letter · Decision Letter 1]

13 Oct 2022

The effect of body image dissatisfaction on goal-directed decision making in a population marked by negative appearance beliefs and disordered eating

PONE-D-22-05219R1

Dear Dr. Seriès,

We’re pleased to inform you that your manuscript has been judged scientifically suitable for publication and will be formally accepted for publication once it meets all outstanding technical requirements.

The reviewer who reviewed your initial submission felt that you had adequately addressed all of their concerns.  Therefore, I am happy to accept your paper for publication.

Kind regards,

Darrell A. Worthy, Ph.D

Academic Editor

PLOS ONE

Additional Editor Comments (optional):

Reviewers' comments:

Reviewer's Responses to Questions

**Comments to the Author**

1. If the authors have adequately addressed your comments raised in a previous round of review and you feel that this manuscript is now acceptable for publication, you may indicate that here to bypass the “Comments to the Author” section, enter your conflict of interest statement in the “Confidential to Editor” section, and submit your "Accept" recommendation.

Reviewer #1: All comments have been addressed

2. Is the manuscript technically sound, and do the data support the conclusions?

Reviewer #1: Yes

3. Has the statistical analysis been performed appropriately and rigorously? 

Reviewer #1: Yes

4. Have the authors made all data underlying the findings in their manuscript fully available?

Reviewer #1: Yes

5. Is the manuscript presented in an intelligible fashion and written in standard English?

Reviewer #1: Yes

6. Review Comments to the Author

Reviewer #1: Thank you for your work with my comments. I am pleased to see the manuscript is now looking well. Good luck with publication.

7. PLOS authors have the option to publish the peer review history of their article (what does this mean?). If published, this will include your full peer review and any attached files.

Reviewer #1: No

---

## [Editor Report · Acceptance letter]

11 Nov 2022

PONE-D-22-05219R1 

The effect of body image dissatisfaction on goal-directed decision making in a population marked by negative appearance beliefs and disordered eating 

Dear Dr. Seriès:

I'm pleased to inform you that your manuscript has been deemed suitable for publication in PLOS ONE. Congratulations! Your manuscript is now with our production department. 

Kind regards, 

on behalf of

Dr. Kamila Czepczor-Bernat 

Guest Editor

PLOS ONE